# Single housing but not changes in group composition causes stress-related immunomodulations in horses

Sonja Schmucker[1]*, Vanessa Preisler[1], Isabell Marr[2], Konstanze Krüger[2], Volker Stefanski[1]

1 Behavioral Physiology of Livestock, Institute of Animal Science, University of Hohenheim, Stuttgart, Germany, 2 Equine Economics, Faculty Agriculture, Economics and Management, Nuertingen-Geislingen University, Nuertingen, Germany

* sonja.schmucker@uni-hohenheim.de

**Data Availability Statement:** The data underlying the results presented in the study are available

## Abstract

Domestic horses are currently often subject to management practices that can entail social stressors, which in turn can negatively influence immunocompetence and disease susceptibility.

The present study therefore aimed to characterize the number of various blood leukocyte subsets in horses, focusing on two potentially stressful housing environments: changes in group composition and relocation to individual stabling. Immune measurements were conducted before as well as one and eight days after changes were made. They were complemented by an assessment of plasma cortisol concentrations as well as behavioral observations. One and eight days after relocation to single housing, the mean numbers of eosinophils, T helper cells and cytotoxic T cells decreased by up to 31%, 20% and 22% respectively, whereas the mean numbers of neutrophils increased by 25%. In contrast, one and eight days after changes in group composition not only the mean number of neutrophils, but also of monocytes, T helper cells and cytotoxic T cells increased by up to 24%, 17%, 9%, and 15% respectively. In consequence, an increase in the neutrophil-to-lymphocyte ratio indicating stress-induced immune modulation was found after relocation to single housing, but not after changes in group composition. The changes in leukocyte numbers after relocation to single housing were accompanied by a transient increase in cortisol concentrations after one day and the occurrence of disturbed behavior patterns one week after change in housing condition. In contrast, changes in group composition did not result in an increase of cortisol concentrations or in an increase of aggressive interactions. The results strongly indicate that individual stabling is an intense stressor leading to acute and lasting alterations in blood counts of various leukocyte types. The study highlights a probable negative impact of single housing on welfare and health of horses and an advantage of group housing systems in view of immunocompetence.

from Zenodo (https://doi.org/10.5281/zenodo.6359830).

**Funding:** The study was funded by the Ministry of Food, Rural Affairs and Consumer Protection of the State of Baden-Württemberg, Germany. The funders had no role in study design, data collection and analysis, decision to publish, or preparation of the manuscript.

**Competing interests:** The authors have declared that no competing interests exist.

## Introduction

Stress-related modulations in number and function of immune cells mediated by stress hormones are well-documented for many species, including horses [1–3]. In this regard, immunomodulations such as higher numbers of blood neutrophils and lower numbers of blood lymphocytes, in particular T cells, are well-accepted indicators of a stress response [1, 4]. Changes in leukocyte numbers induced by acute stress responses seem to display an evolutionary process of immune enhancement and recruitment at times when most needed, whereas chronic stress might have detrimental outcomes for the health of an organism due to a dysregulation of immune functions, and thus reduced immunocompetence [5, 6]. As such, assessing factors leading to stress responses and influencing the immunocompetence of animals under domestic conditions is of high importance in regard to their health and welfare.

Social factors are potent in inducting stress responses, and in particular chronic stress, which is why they are known as social stressor [7, 8]. Within animal management practices, social interactions are often influenced by the housing conditions provided. In this regard, horses are regularly housed by individual stabling [9, 10] usually without full contact to conspecifics leading to social isolation and compromising the natural needs of horses [11]. Naturally, horses are free-roaming, pasture-grazing herd animals with complex social relationships [12]. As a consequence, single housing is already known to lead to stress responses in horses as indicated by changes in heart rate [13], high prevalence of abnormal behaviors [13–15] and alterations in glucocorticoid levels [16] as well as hypothalamus-pituitary-adrenal (HPA) axis function [15]. In addition, individually stabled horses may not adapt to these housing conditions but become unresponsive to environmental stimuli and apathetic the longer the individual stabling lasts [17].

As an alternative to single housing, group housing systems are becoming more widely used [18]. However, social contact between individuals always involves the potential of agonistic interactions, which can evoke social stress [7, 19]. The occurrence of agonistic encounters depends on various factors such as a horse´s individual experiences, personality, or the composition of the social group in regard to age, sex, and stability [19, 20]. Aggressive interactions often occur during hierarchy establishment after disruption of the existing social structure, for example by changes in group composition [19, 20]. However, due to management requirements, regroupings often have to be applied. Despite differences in the nature of the stressor, both individual stabling as well as group housing might therefore lead to social stress, and in consequence to an impaired immunocompetence of the horses.

Interestingly, in contrast to the effects of transport, exercise, and competition [21–24], immune parameters of horses in the context of housing conditions were only investigated in one study so far. Lesimple et al. (2020) found that blood cell counts of leukocytes fall outside the normal range within a high percentage of single-housed horses [25]. No further studies have so far examined possible immune-modulating effects of housing conditions on horses. The present study therefore aimed at characterizing the number of blood immune cells in horses in more detail under two potentially stressful housing environments: changes in group composition and relocation to individual stabling. Determination of immune measures were complemented by assessing plasma cortisol concentrations, aggressive interactions, and stereotypy-related behavior, all of which are indicative of a stress response [26]. It was hypothesized that the two changes in housing environment would evoke an acute stress response and a change in leukocyte numbers. Single housing, in addition, should also have a high potential of inducing chronic stress, thus leading to immune modulations lasting longer than 24 hours.

## Materials and methods

All procedures used in this study adhered to current animal ethics and animal care guidelines, and were approved by the local authority´s Animal Ethics Committee (Regional Administrative Authority Tübingen; permit number: 35/9185.81-4/).

### Animals and housing

A total of 15 two and three-year-old German Warmblood castrated male horses (*Equus caballus*), raised and housed at the state stud farm Marbach (Gomadingen, Germany) were included in the study. The horses were housed in one social group on pasture (approximately 2.2 ha) with ad libitum access to grass and water for at least 6 weeks prior to any experimental changes in housing conditions. Changes in housing conditions encompassed two treatments, which were conducted successively: First the fission of the established social group and second the relocation to single housing (Fig 1). For group fission, horses were subdivided into two groups of $N = 7$ and $N = 8$ and relocated to two different paddocks (approximately 2 ha each), which were unknown to the horses and did not allow visual contact between the groups. Both paddocks were comprised of pasture and provided ad libitum access to water and grass. For group composition, horses within similar rank classes (high, middle, low; determination see below) were evenly assigned to the two groups in a random manner. Horses were kept for eight days in this new group composition. Thereafter the groups were fused again to original setting. After a further period of eight weeks, in which the overall group size decreased to $N = 12$ due to selling of individual horses, the remaining animals were relocated to single-housing conditions in a stable. Individual boxes were 3.2 x 3.5 m and enabled sight as well as tactile contact with neighboring horses through barred windows. Water and hay were provided ad libitum and horses received grain three times a day meeting each horse's individual needs. During the first week after relocation to the stable, free movement of the horses was enabled in groups of $N = 6$ within an indoor area of the stable for 30 minutes per day. From the second week onwards, the horses were trained by lunging.

### Blood sampling by vena jugularis puncture

For assessment of immunological effects of the above-named treatments, blood samples were collected from all animals at three times before (-7 days, -6 days and immediately before) as

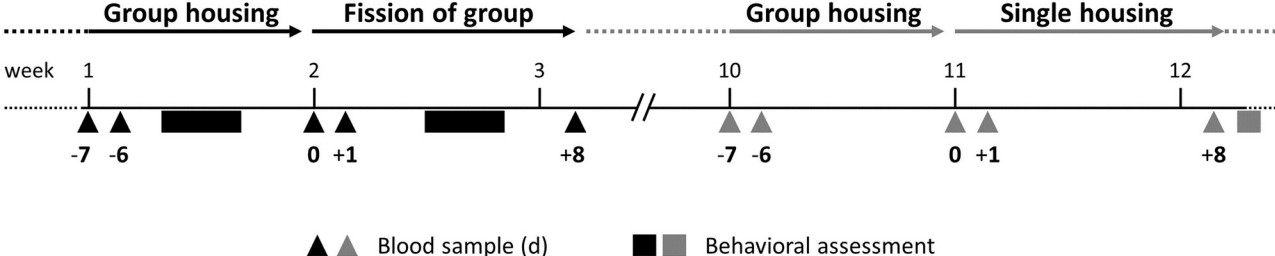

**Fig 1. Scheme of study design.** Two and three-year-old German Warmblood castrated male horses ($N = 15$) were housed in one stable social group on pasture at the start of the study. The group was subdivided and horses were kept for eight days (d) in two groups ($N = 7$ and $N = 8$) in different paddocks comprised of pasture. After fusion, the horse were kept again as one stable social group for a period of eight weeks. Afterwards, the horses were relocated to single housing conditions in a stable ($N = 12$). Blood samples for analysis of immune cell numbers and cortisol concentrations were collected from all animals while in the stable group (-7 days, -6 days, immediately before change = 0) and 1 day and 8 days after both changes in housing conditions. Behavioral assessment of social rank and social interactions was conducted during group housing, and stereotypy-related behavior was assessed after relocation to single housing.

well as one day and eight days after both changes in housing conditions (Fig 1). The horses had already been habituated to being guided into an enclosure at the paddock in case of management interventions. To obtain blood while on paddock, horses were guided into the enclosure by provision of a small amount of grain and venipuncture of the jugular vein was carried out by the veterinarians of the stud farm. While in single housing, blood sampling was carried out in the individual boxes. Animals were sampled between 0800–1000h in the same order each time, and mean sampling time per horse was <3 min. Blood was collected into lithium heparin tubes and K3 EDTA tubes (both Sarstedt, Nürnbrecht, Germany).

## Characterization and enumeration of leukocyte subsets in blood

Relative numbers of T helper (TH) cells, cytotoxic T cells (CTL), neutrophils, eosinophils and monocytes were quantified in heparinized whole blood by flow cytometric analysis after two-color immunofluorescent antibody (Ab) staining. Therefore, cells were labelled with the monoclonal Abs: mouse anti-horse CD5 (conjugated to RPE, clone CVS5; Bio-Rad Laboratories, Feldkirchen, Germany) and either mouse anti-horse CD4 (clone CVS4 conjugated to FITC; Biozol, Eching, Germany) or mouse anti-horse CD8α (clone CVS8 conjugated to FITC; Biozol) by a standard protocol already described in [27]. In brief, 20µL of equine whole blood was incubated with or without well-titrated concentrations of the above-named Abs for 15 min at room temperature (RT). Subsequently, erythrocytes were lysed and leukocytes were fixed using BD FACS Lysing Solution (BD Biosciences, Heidelberg, Germany) according to the manufacturer´s instructions. Ab-labelled samples were stored at 4˚C until flow cytometric analysis of at least 10,000 cells on a BD FACSCalibur™ (BD Biosciences), using the software BD CellQuest Pro 6 (BD Biosciences). An unlabeled control was included for determination of unspecific binding of the used Abs and compensation was performed by single stained controls. The gating strategy is illustrated in Fig 2. Lymphocytes, monocytes and granulocytes were differentiated based on light scatter characteristics (size and granularity). Granulocytes were further divided within the unlabeled control into neutrophils and eosinophils by the well-known autofluorescent properties of eosinophils [28]. TH cells ($CD5^+CD4^+$) and CTL ($CD5^+CD8\alpha^+$) were discriminated among the lymphocyte population within the corresponding Ab-labelled samples. Total leukocyte counts were analyzed in EDTA blood using an automated hematology analyzer (pocH 100-iV Diff; Sysmex, Norderstedt, Germany). By combining flow cytometric analysis of relative cell numbers and hematologic total leukocyte count, the absolute cell number per µL whole blood of each analyzed immune cell type was calculated. The neutrophil-to-lymphocyte ratio (N:L ratio) was determined by dividing the number of neutrophils by the number of lymphocytes.

## Determination of plasma cortisol concentration

Plasma was obtained from the heparinized blood samples taken immediately before (point in time 0; Fig 1), one day after and eight days after either treatment by centrifugation at 1000×$g$ for 10 minutes and stored at −80˚C until further processing. Cortisol concentrations in these plasma samples was determined in duplicate by radioimmunoassay (RIA) after extraction with ethyl acetate. Plasma was diluted 1:5 in distilled water, and ethyl acetate (AppliChem, Darmstadt, Germany) was added. After incubation and freezing the aqueous phase, the non-frozen solvent supernatant was collected and evaporated in a vacuum dryer for 40 minutes at 55˚C. Extracts were resuspended in phosphate buffer and analyzed in the RIA as described by Engert et al. (2017) [29] with the few following modifications. In brief, an in-house prepared polyclonal rabbit-anti-cortisol antiserum at a final dilution of 1:51,000 was used in combination with [1,2,6,7-$^3$H]-cortisol (78,3 Ci/mmol; PerkinElmer, Boston, MA, USA) as a tracer. The

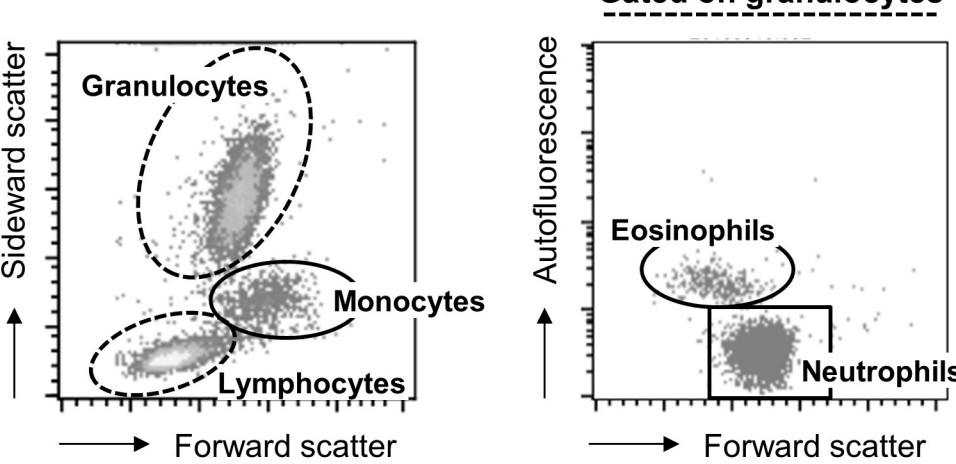

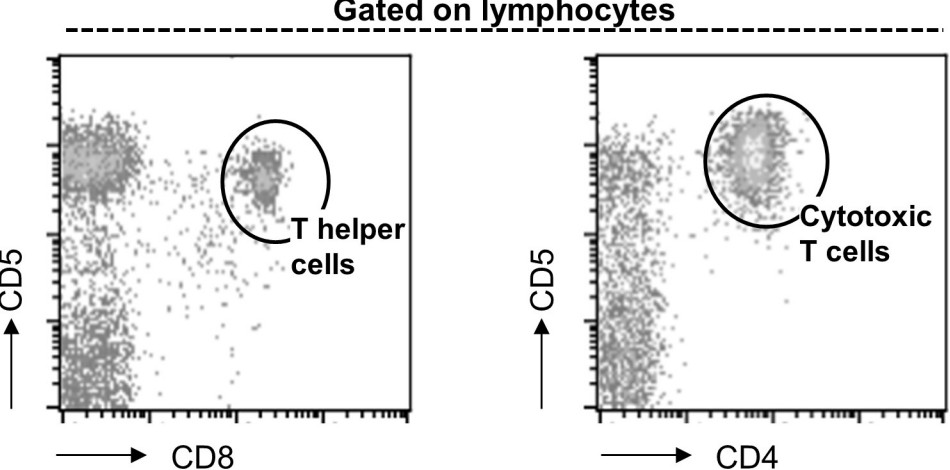

**Fig 2. Gating strategy for flow cytometric discrimination of equine leukocyte populations in whole blood.**
Immune cells were left unlabeled or labeled with an antibody specific for equine CD5 in combination with an antibody
either specific for equine CD8 or for equine CD4 and analyzed on a flow cytometer.

polyclonal antiserum revealed a cross-reactivity of 12.1% for corticosterone, 1.4% for proges-
terone, 6.8% for desoxy-corticosterone and 1.2% for aldosterone. After separation of bound/
free by centrifugation with dextran-coated charcoal (0.05% Dextran 70, Carl Roth, Karlsruhe,
Germany; 0.5% Norit A, Serva Electrophoresis, Heidelberg, Germany) at $2,000 \times g$ for 20 min
at 4˚C, supernatants were transferred to 5 mL Ultima-Gold (PerkinElmer) and radioactivity
was determined (Tri-Carb 2800 TR, PerkinElmer). The standard curve was prepared in phos-
phate buffer and covered a range of 0.1 ng/mL to 10 ng/mL cortisol (Sigma Aldrich, St. Louis,
MO, USA). To determine procedural losses during extraction, $^3$H-cortisol-spiked samples
($[1,2,6,7-^3$H]-cortisol; 78,3 Ci/mmol; PerkinElmer) were integrated into each sample batch
and results were compensated for respective recovery rates (in average 88.8%). Accuracy of the
RIA, determined with cortisol-spiked controls (50 and 75 ng/ml; Sigma Aldrich), revealed a
mean relative error of 131.2% and 111.4%, respectively. Intra-assay variability for a biological

sample was 7.7%, and inter-assay variability for biological samples with high and low endogenous cortisol were 9.4% and 6.7%, respectively.

## Behavioral assessments

For evaluation of probable stress responses by the horses after either change in group composition or relocation to individual stabling behavioral measurements by different ethograms were applied.

As stress burden within group housing of horses largely depends on aggressive interactions and also might be reflected by changes in social hierarchy [7, 19, 20] behavioral assessments were conducted in order to determine social hierarchy and social interactions before and after changes in group composition. Hierarchy was determined by calculation of the average dominance index (ADI) [30]. Behaviors for determination of ADI comprised the agonistic (aggressive and submissive) behaviors "leading", "following", "approach", "bite", "threat to bite", "kick", "threat to kick", "chase", and "retreat" based on the ethogram of McDonnell (2003) and McDonnell and Haviland (1995) [31, 32] including the definition of sender and receiver of each sampled behavior. The winner of an agonistic interaction was defined as the horse causing a retreat by the other horse. Wins of every pairing were counted and ADI of individual horses was calculated using the formula:

$$\text{ADI} = 1/N \sum_j \left[ x_{ij} / \left( x_{ij} + x_{ji} \right) \right], \tag{1}$$

where $N$ = number of interacting partners, $x_{ij}$ = numbers of wins of individual i versus individual j, $x_{ji}$ = numbers of wins of individual j versus individual i. In addition, the number of social interactions between horses before and after fission of the stable group was analyzed based on the above-named agonistic behaviors and extended by the affiliative behaviors "grooming" "approach", "mutual grooming", "mutual grazing", "play" and "approach without retreat" based on the ethogram of McDonnell (2003) [31]. Social interactions were assigned to either the categories "sent aggressive interactions", "received aggressive interactions"(comprising threats and aggressive encounters), "sent affiliative interactions", or "received affiliative interactions"per individual horse and calculated as mean per horse and hour. All observations comprising social interactions and determination of hierarchy were conducted for 12 hours in total before fission of the group as well as 12 hours in total for each group after fission of the group by two experienced observers. Observations were evenly distributed over various days and times of day. Behavioral assessment was done by observing all individuals simultaneously for displaying one of the above-named agonistic and affiliative behaviors (behavioral sampling) and recording each occurrence as well as sender and receiver of the particular behaviors (continuous recording) [33].

The occurrence of stereotypic behaviors is considered as indicator of a potential stress burden within stabled horses [13–15], thus in the present study the development of disturbed behavior patterns was characterized after relocation to single housing. One week after relocation to single housing, horses were individually observed for any occurrence of stereotypy-related behavior (Table 1). Observations were made once per horse for 2 hours by continuous recording and behavioral sampling of the behaviors named in Table 1 by focusing on five to six animals in parallel (focal sampling) [33] by one experienced observer.

## Data handling and statistical analysis

All statistical data analyses were performed with R version 3.6.1 and 4.1.2 (R Foundation for Statistical Computing, Vienna, Austria) using linear mixed models with the lme4 [38] and

**Table 1. Ethogram for stereotypy-related behaviors and behavior patterns associated with frustration or stress response in stabled horses.**

| Category | Behavior | Description | Causal factor[§] |
|---|---|---|---|
| Locomotion | Box walking | The horse repeatedly walks in its box in complete 360° or sequences of 2 x 180° circles | Appetitive locomotor behavior |
| | Weaving (partly) | Horse shakes its head from one side to the other while additionally shifting its weight from one forehand to the other (also in part) | Frustrated behavior (appetitive locomotor or social) |
| Oral | Manipulation of manger or drinking pod | Manipulation of manger or drinking pod with lips, teeth or tongue without feed intake | Appetitive feeding behavior |
| | Wood chewing / licking box wall | Manipulation of wall of box with lips, teeth or tongue with or without uptake of wood | Appetitive feeding behavior |
| | Bar-biting | Manipulation of bars (window to neighboring horse box) with teeth | Appetitive feeding or social behavior |
| | Cribbing / wind sucking | The horse seizes its top incisor teeth on a fixed object or arches its neck while drawing air into the cranial esophagus by tensing its cervical muscles (also in part) | Stress-related coping or visceral discomfort (gastric acidity) |
| Social interaction | Paw / paw at box side | A front leg is lifted and the toe dragged against the ground or the wall of the box in a digging motion | Social interaction / social frustration |
| | Stamp | A leg is raised and fast and firmly lowered to the ground | Usually seen during ritualized interactions |
| | Neighing | The horse neighs | Social communication |
| Apathetic | Head to back side of box | The horse stands with all four legs on the ground with its head to the backside of the box without dozing or sleeping, exerting a state of immobility. Horse may also show withdrawn posture [34] | Stress-related coping |
| | Standing flat to wall | The horse stands with all four legs on the ground and its body parallel to the wall without dozing, sleeping or being alert, exerting a state of immobility | Stress-related coping |

[§] based on [17, 32, 34–37]

lmerTest [39] package. The following model was used:

$$Y_{ijkl} = \mu + \alpha_i + \beta_j + (\alpha\beta)_{ij} + \gamma_k + \delta_l + \varepsilon_{ijkl}, \tag{2}$$

where for analysis of immune cell numbers and plasma cortisol concentration $Y_{ijkl}$ = response variable, $\mu$ = overall mean, $\alpha_i$ = effect of treatment (fission of group or relocation to single housing; fixed), $\beta_j$ = effect of point in time (fixed; prior to, 24h after and 8 day after change in housing condition; phrase shortened to "time" within display of results), $(\alpha\beta)_{ij}$ = the interaction between treatment and point in time (fixed; phrase shortened to "treatment×time" within display of results), $\gamma_k$ = individual animal (random), $\delta_l$ = rank class (fixed; high, middle, low; shortened to "rank" within display of results), and $\varepsilon_{ijkl}$ = residual error.

A similar model was used for the analysis of social interactions, where $Y_{ijkl}$ = response variable, $\mu$ = overall mean, $\alpha_i$ = effect of rank class (high, middle, low; fixed; shortened to "rank" within display of results), $\beta_j$ = effect of point in time (prior to and post changes in housing condition; fixed; phrase shortened to "time" within display of results), $(\alpha\beta)_{ij}$ = the interaction between rank class and point in time (fixed; phrase shortened to "time×rank" within display of results), $\gamma_k$ = individual animal (random), $\delta_l$ = group (A or B; fixed), and $\varepsilon_{ijkl}$ = residual error. Variance components were estimated using the restricted maximum likelihood (REML) method and degrees of freedom were determined by the Kenward-Roger method. Multiple pairwise post-hoc testing adjusted by the Tukey method was performed using the lsmeans package [40]. Residuals were tested for normal distribution and homogeneous error variance via the graphical check of residual plots [41]. Logarithmic transformations were used to stabilize variance and to meet the distribution assumption for the datasets comprising the N:L ratio, cortisol concentrations, and sent aggressive interactions. Statistical significance was

declared at $P < 0.05$ and statistical trends at $P < 0.1$. The single main effects of treatment, time, or rank were only considered in case of non-significant interactions. Results are presented as the LSmeans and SE of the untransformed data.

Due to medical treatments, blood samples of individual horses had to be excluded from immunological and cortisol analyses at particular points in time. Moreover, when the groups were separated, a tree fell in the paddock of one of the two groups due to stormy weather the night before day eight after group separation, resulting in clear stress-related behavior of the horses (abnormal high running in the paddock, high alertness, high frequency in neighing). In consequence, the affected horses had to be excluded from analysis for this point in time. The resulting data set during the treatment "fission of group" included $N = 13$ animals for the point in time "prior to treatment" (integrating the points in time: −7d, −6d and 0 in case of blood immune cell numbers and comprising the point in time 0 for cortisol concentrations; Fig 1) as well as 1 d after start of treatment and $N = 5$ for 8 d after start of treatment. For the treatment "single housing" the data set comprised $N = 12$ animals for the point in time "prior to treatment" as well as 1 d after start of treatment and $N = 11$ for 8 d after start of treatment. For statistical analysis of social interactions during group housing $N = 15$ horses were included with $N = 5$ high ranking horses, $N = 6$ middle ranking horses and $N = 4$ low ranking horses. Analysis of stereotypy-related behavior one week after relocation to individual stabling was performed with $N = 11$ horses.

## Results

### Numbers of circulating immune cells considerably change after relocation to single housing

An interactive effect of time×treatment was found in the blood count of all investigated leukocyte types (monocytes: $F(2,92.28) = 11.55$, $P < 0.001$; eosinophils: $F(2,92.38) = 4.18$, $P = 0.018$; neutrophils: $F(2,92.62) = 3.45$, $P = 0.036$; TH cells: $F(2,92.17) = 37.56$, $P < 0.001$; CTL: $F(2,92.15) = 26.97$, $P < 0.001$; N:L ratio: $F(2,93.38) = 18.49$, $P < 0.001$), revealing differences in immunomodulations following the two compared changes in housing conditions (Fig 3A and 3B).

After fission of the stable group, the numbers of almost all investigated immune cell types increased, except for eosinophils (1d vs. 0: $t(92.0) = 1.072$, $P = 0.891$; 8d vs. 0: $t(93.0) = 0.202$, $P = 1.0$; 1d vs. 8d: $t(92.8) = 0.817$, $P = 0.964$). The increase in mean numbers / µL blood comprised up to 24% for neutrophils, 17% for monocytes, 9% for T helper cells, and 15% for cytotoxic T cells. In detail, a higher number of monocytes ($t(92.0) = 2.872$, $P = 0.055$), TH cells ($t(92.0) = 3.487$, $P = 0.01$), and CTL ($t(92.0) = 3.727$, $P = 0.004$) was found 1d after fission of the group. This increase in cell counts lasted or tended to last until 8d after the change in group composition (monocytes: $t(92.7) = 3.008$, $P = 0.039$; TH cells: $t(92.4) = 2.646$, $P = 0.097$; CTL: $t(92.4) = 2.822$, $P = 0.063$). The number of neutrophils (1d vs. 0: $t(92.2) = 2.271$, $P = 0.217$; 8d vs. 0: $t(93.5) = 3.8$, $P = 0.003$) increased at 8d after change in group composition. The N:L ratio did not change after fission of the group (Fig 3C; 1d vs. 0: $t(93.0) = 0.225$, $P = 1.00$; 8d vs. 0: $t(93.9) = 1.168$, $P = 0.851$; 1d vs. 8d: $t(93.7) = 0.931$, $P = 0.938$).

In contrast to group fission, relocation to single housing was associated with different changes in the numbers of the investigated immune cell types in blood. Whereas the mean numbers of neutrophils increased by 25%, the mean numbers of monocytes, eosinophils, T helper cells and cytotoxic T cells decreased by up to 16%, 31%, 20% and 22%, respectively. Statistical analyses revealed that the numbers of monocytes ($t(92.1) = 3.725$, $P = 0.004$), eosinophils ($t(92.1) = 4.388$, $P < 0.001$), TH cells ($t(92.0) = 8.169$, $P < 0.001$) and CTL ($t(92.0) = 5.693$, $P < 0.001$) were decreased at 1d of single housing. The number of eosinophils ($t(92.1) = $

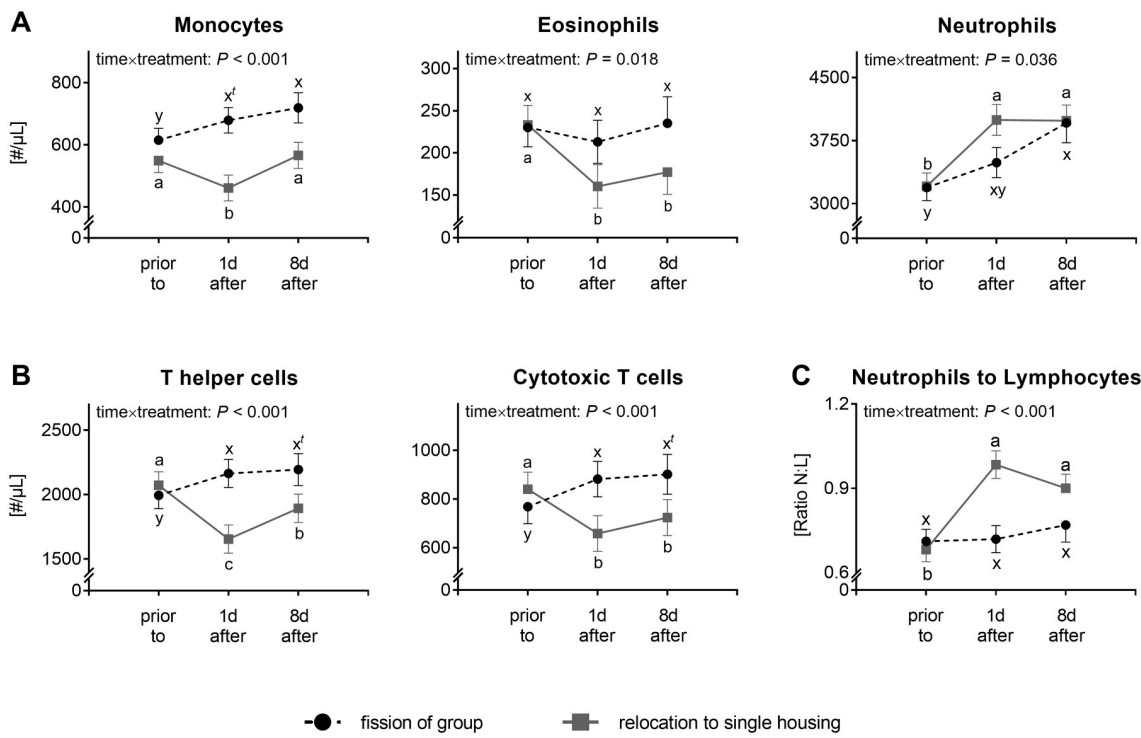

**Fig 3. Numbers of various leukocytes types in blood of horses exposed to either changes in group composition or relocation to single housing.** (A) Numbers of peripheral innate immune cells; (B) Numbers of peripheral T cell subsets; (C) Ratio of the numbers of neutrophils to lymphocytes (N:L ratio) in blood. Immune cell numbers are depicted as LSmeans ± SE (only shown in one direction in case of overlapping SE; $N$ = 5–13) and results of linear mixed model analyses are shown within the graph sparing the results of effect of time and effect of treatment if time×treatment interaction with $P < 0.05$. Different lowercases depict significant differences ($P < 0.05$) or tendencies ($P < 0.1$; depicted by additional $t$) between point in times within particular treatment (fission of group: x-y; relocation to single housing: a-c).

3.238, $P$ = 0.02) and CTL ($t(92.0)$ = 3.543, $P$ = 0.008) still remained low at day 8 after relocation. The number of TH cells increased between 1d and 8d ($t(92.0)$ = 3.755, $P$ = 0.004), but remained at lower levels than before relocation ($t(92.0)$ = 3.375, $P$ = 0.013). Monocytes returned to pre-relocation values at day 8 (1d *vs.* 8d: $t(92.0)$ = 3.599, $P$ = 0.007; 0 *vs.* 8d: $t(92.1)$ = 0.722, $P$ = 0.979). In contrast to the leukocyte types named above, the numbers of neutrophils were increased at 1d ($t(92.2)$ = 5.73, $P < 0.001$) and 8d ($t(92.2)$ = 5.450, $P < 0.001$) after relocation to single housing. The alterations resulted in an increased N:L ratio at 1d ($t(93.0)$ = 8.965, $P < 0.001$) and 8d ($t(93.1)$ = 6.244, $P < 0.001$) after relocation to single housing (Fig 3C).

No effect of rank was found for the number of any of the investigated circulating leukocyte types nor for the N:L ratio (monocytes: $F(2,28.06)$ = 0.28, $P$ = 0.762; eosinophils: $F(2,27.42)$ = 2.201, $P$ = 0.13; neutrophils: $F(2,26.06)$ = 0.06, $P$ = 0.942; TH cells: $F(2,28.68)$ = 0.86, $P$ = 0.434; CTL: $F(2,28.82)$ = 0.55, $P$ = 0.58; N:L ratio: $F(2,11.96)$ = 1.76, $P$ = 0.214).

## Concentrations of plasma cortisol

After fission of the stable group, no changes in cortisol concentrations occurred (Fig 4; 1d *vs.* 0: $t(44.1)$ = 1.42, $P$ = 0.715; 8d *vs.* 0: $t(44.5)$ = 0.44, $P$ = 0.998; 1d *vs.* 8d: $t(45.3)$ = 1.47, $P$ = 0.687). In contrast, relocation to single housing was associated with an increase in cortisol

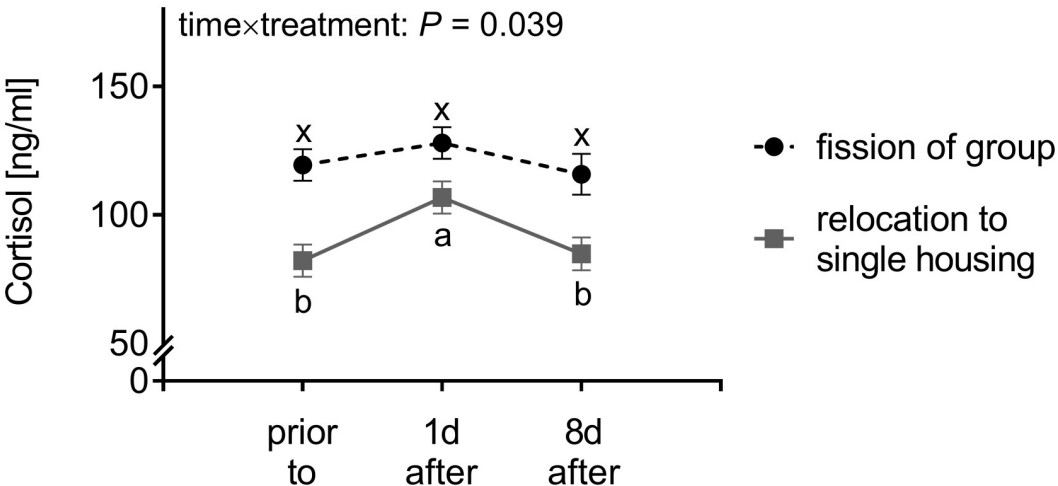

**Fig 4. Cortisol concentration in blood of horses raises upon relocation to single housing but not after fission of the stable group.** Plasma cortisol concentrations of horses after relocation to single housing and fission of the stable group within point in time are depicted as LSmeans ± SE (*N* = 5–13) and results of linear mixed model analyses are shown within the graph sparing the results of effect of time and effect of treatment if time×treatment interaction with *P* < 0.05. Different lowercases depict significant differences (*P* < 0.05) between point in times within particular treatment (fission of group: x-y; relocation to single housing: a-b).

at 1d after change in housing conditions ($t(44.1)$ = 5.08, $P$ < 0.001), causing an interactive effect of time×treatment ($F(2, 44.55)$ = 3.49, $P$ = 0.039). Concentrations of plasma cortisol returned to baseline levels 8d after relocation to single housing (8d *vs.* 0: $t(44.1)$ = 0.77, $P$ = 0.971; 1d *vs.* 8d: $t(44.1)$ = 4.17, $P$ = 0.002). No effect of rank was found for the concentration of plasma cortisol ($F(2, 23.98)$ = 0.73, $P$ = 0.764).

### Fission of the stable group induces higher numbers of affiliative interactions

Fission of the group did not change the rank class of any individual horses in the newly subdivided groups (Fig 5A).

Neither the number of sent nor of received aggressive interactions increased after fission of the group (Fig 5B). For received aggressive interactions no effect of time×rank ($F(2, 12)$ = 0.24, $P$ = 0.791) and no effect of time ($F(1, 12)$ = 3.16, $P$ = 0.102) was found. An effect of rank ($F(2, 11)$ = 12.62, $P$ = 0.001) showed that low- and middle-ranking horses received a higher number of aggressive encounters than high-ranking horses (low *vs.* high: $t(11)$ = 4.97, $P$ = 0.001, middle *vs.* high: $t(11)$ = 3.11, $P$ = 0.025, low *vs.* middle: $t(11)$ = 2.26, $P$ = 0.105). For sent aggressive interactions a time×rank effect ($F(2, 12)$ = 11.22, $P$ = 0.002) resulted from a decrease in the number of sent aggressive behaviors of low-ranking horses ($t(12.0)$ = 3.95, $P$ = 0.019), which was not seen in middle- ($t(12.0)$ = -1.96, $P$ = 0.416) or high-ranking ($t(12.0)$ = 1.71, $P$ = 0.551) horses. Thus, low-ranking horses sent lower numbers of aggressive behaviors compared to middle- and high-ranking horses after fission of the group (low *vs.* middle: $t(21.4)$ = 6.69, $P$ < 0.001; low *vs.* high: $t(21.4)$ = 7.54, $P$ < 0.001; middle *vs.* high: $t(21.4)$ = 1.23, $P$ = 0.817) whereas there were no differences in sent aggressive interactions between rank classes before fission of the group (low *vs.* middle: $t(21.4)$ = 1.35, $P$ = 0.754; low *vs.* high: $t(21.4)$ = 2.48, $P$ = 0.176; middle *vs.* high: $t(21.4)$ = 1.3, $P$ = 0.781).

In contrast, both sent ($F(1,12)$ = 22.36, $P$ < 0.001) and received ($F(1,12)$ = 17.6, $P$ = 0.001) affiliative interactions were found to increase after fission of the group within all rank classes

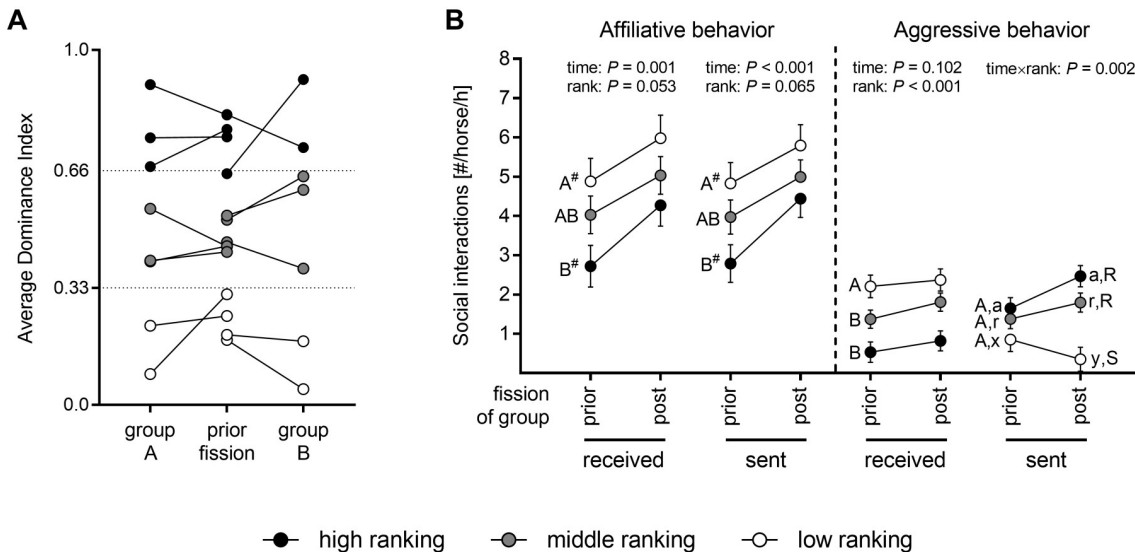

**Fig 5. Fission of a stable group does not change the rank class of individual horses but induces higher numbers of affiliative interactions.** (A) Average dominance index of individual horses before and after fission of a stable group. Individual horses are depicted by circles. ADI of 0–0.33 = low ranking, 0.34–0.65 = middle ranking, 0.66–1.0 = high ranking horses; (B) Number of received and sent affiliative and agonistic interactions per horse and hour before and after fission of a stable group. Interactions are depicted as LSmeans ± SE (in case of overlapping SE only one direction is shown) for each rank class separately. Results of linear mixed model analyses are depicted within the graph sparing result of time×rank interaction if $P > 0.05$, or results of single effects of time and rank if time×rank interaction with $P < 0.05$. In case of missing effect of time×rank interaction: Different uppercase depict significant differences ($P < 0.01$) and different uppercases marked by # depict statistical trend ($P < 0.1$) between rank classes irrespective of point in time. In case of significant effect of time×rank interaction: Different uppercases depict significant differences ($P < 0.001$) of rank at particular point in time (prior to fission: A-B, post fission: R-S) and different lowercases depict significant differences ($P < 0.05$) between points in times of particular rank class (high ranking: a-b, middle ranking: r-s, low ranking: x-y); N = 15; black circles = high ranking horses (N = 5), gray circles = middle ranking horses (N = 6), white circles = low ranking horses (N = 4).

(Fig 5B). In general, rank class tended to be associated with the number of sent ($F(2,11) = 3.53$, $P = 0.065$) and received ($F(2,11) = 3.89$, $P = 0.053$) affiliative interactions, with low-ranking horses receiving ($t(11) = 2.77$, $P = 0.044$) and tending to send ($t(11) = 2.651$, $P = 0.055$) more affiliative interactions than high-ranking horses. No significant effect of group was found for the number of sent ($F(1, 11) = 1.47$, $P = 0.252$) and received affiliative ($F(1, 11) = 0.41$, $P = 0.534$) or sent ($F(1, 11) = 1.27$, $P = 0.284$) and received agonistic interactions ($F(1, 11) = 1.36$, $P = 0.268$; Fig 5B).

## Horses show pre-stages of stereotypy-related behavior shortly after relocation to single housing

The analysis of stereotypy-related behavior patterns showed that all horses exhibited non-appropriate locomotory behaviors in regard to stabling in single boxes one week after relocation to single housing (Fig 6). One horse was found weaving. In total, 10 out of 11 horses showed oral manipulation of non-adequate objects not related to food uptake and one horses showed pre-stages of cribbing. Behavior patterns related to social interactions and contact deprivation such as neighing and stamping as well as behaviors related to frustration such as pawing were shown by 9 out of 11 horses during the 2 h behavioral observation period. Apathetic behavior was found in 9 out of 11 horses.

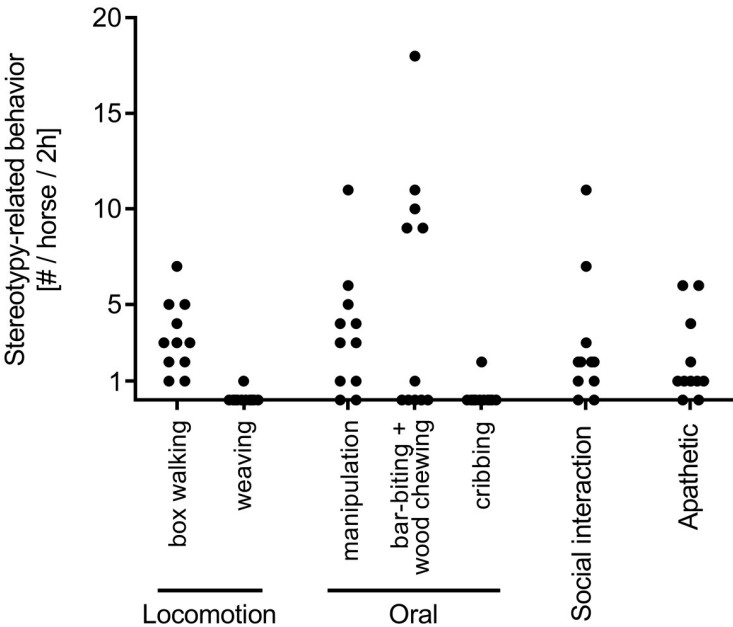

**Fig 6. Horses show stereotypy-related behavior one week after relocation to single housing.** Number of stereotypy-related behavior or its pre-stages per individual horse and category are depicted by black circles.

## Discussion

The results of the present study show that relocation to single housing in a stable led to acute stress-induced immune modulations, whereas changes in the group composition did not. As hypothesized, the immune-modulations following relocation to single housing partly lasted for at least one week and were accompanied by occurrences of disturbed and stereotypy-related behavior patterns.

In general, cell numbers of the various immune cell types and cortisol concentrations of the horses were well within ranges found by other studies [42, 43]. Consistent with the initial hypothesis, relocation of the horses to single housing led to changes in the numbers of all investigated immune cell types. The number of eosinophils, monocytes and T cells declined, whereas the number of neutrophils increased resulting in an increased N:L ratio. This pattern of change resembles the well-known picture of an immunomodulation induced by acute social stress [3, 6, 44]. These immunological changes were accompanied by a short-term increase in cortisol concentrations, which is known to induce trafficking events of particular lymphocytes [5]. Although cortisol concentrations returned to baseline level after 8 days, the alterations in most immune cell numbers persisted, pointing to a longer-lasting effect on the immune system of the horses. In line with the results of the present study, Lesimple et al. (2020) found cell counts out of normal ranges in 12.5%–91.6% of horses stabled for at least one year [25].

The lasting effect of immune alterations found in the present study and in the study of Lesimple et al. (2020) [25] indicate that individual stabling of horses has the potential of caus-ing chronic immune dysregulation. The stress-related alterations in immune cell numbers might also be indicative of an impairment of immune function in general as is reflected in many studies with various mammalian species, including horses [7, 22, 42]. A thereby resulting decrease in immunocompetence might increase disease susceptibility of the horses and thus impair their health and welfare.

The results of the present study also show that alterations in immune parameters might be better indicators of chronic stress responses than glucocorticoid concentrations. A continuous secretion of glucocorticoids due to a chronic stress-response may lead to a desensitization of the HPA axis, and in consequence, to glucocorticoid-resistance of the animal. This was also suggested as an underlying mechanism in first-time stabled horses showing low response of the HPA-axis in a study by Visser et al. (2008) [15]. Moreover, Pawluski et al. (2017) showed that horses with compromised welfare, scored by inclusion of various health-related parameters, had even lower concentrations of plasma cortisol as well as fecal cortisol metabolites than conspecifics with normal welfare scores [45]. It is worth noting that Pawluski et al. (2017) also characterized abnormal neutrophil counts in the welfare-compromised animals [45]. In addition, Popescu and Diugan (2017) found a negative correlation between the N:L ratio and individual welfare scores in working and breeding horses [4], highlighting the suitability of immune parameters for assessment of chronic stress responses.

The assumption that relocation to individual stabling represents a strong stressor for horses is also underlined by the occurrence of stereotypy- and disturbance-associated behavior patterns as early as one week after relocation. Such behavior patterns are well-accepted indicators of a stress response and of poor welfare [26]. In addition, an accompanying investigation with the horses of the present study found changes in sensory and motor laterality after relocation to single housing also pointing to an acute and chronic stress response [46]. Whether social isolation and missing locomotor and grazing options contribute equally to the stress burden of individual-stabled horses cannot be distinguished in the present study. However, studies investigating the occurrence of disturbed behavior or other welfare indicators in horses stabled either individually, pairwise, or in groups showed a higher burden of stress in the single-stabled horses [15], pointing to a strong effect of social deprivation. In addition, offering individual-stabled horses the possibility of movement and exploration by solitary free time in an outdoor paddock, but without offering social interactions with conspecifics, did not lead to alterations in immune parameters [25]. Further investigating immune parameters in the context of certain management practices (such as continuous versus interrupted stabling, pairwise versus single housing, or daily access to other horses) could further clarify the impact of social isolation on horses´ immunocompetence, and thus contribute to improvements in regard to health and welfare.

In contrast to relocation to single housing, fission of the group and the resulting change in group composition did not lead to any modulation of immune parameters or changes in plasma cortisol concentrations indicative of a stress response. Although the neutrophil blood counts of the horses of the present study increased in the week after fission of the group, all other immune cell types also did. Thus, this effect most probably does not reflect a stress response. A more likely explanation is that the relocation to an unknown paddock increased the novelty-associated locomotor activity of the horses. Moderate exercise is known to enhance recirculation and to increase the number of leukocytes in the blood of humans and rodents as well [47, 48].

Within group housing, social stress usually results from agonistic interactions between individuals [7]. In the present study, no increase in the number of aggressive interactions, but instead a rise in affiliative interactions was found. Moreover, the social rank of the individual horses did not change even after the group was divided. As agonistic interactions among horses of a stable group are very rare and subtle once a hierarchy has been established [49], the putative increase in rank of one horse after fission of the group most probably reflects the difficulties of observing rank-specific behavior patterns without experimentally creating competitive situations. In summary, the horses of the present study were found to consolidate their social bond by higher numbers of affiliative encounters rather than by fighting over the re-

establishment of hierarchy. A similar effect was observed by Bourjade et al. (2008) after introducing adults into groups formed by 1- and 2-year-old horses [50]. Affiliative encounters are known to improve the stability of social groups [51] and have already been suggested as suitable parameters for assessment of social bonds within horse groups [52]. In addition, a strong social cohesion but also a migration of individual horses or groups of horses also seem to be common among bachelor bands under natural conditions [12]. Indeed, fission-and-fusion systems seem to be the norm and not the exception [12]. Moreover, the horses of the present study were well-experienced when it came to interactions with other horses, a factor which was found to contribute to low numbers of aggressive encounters [20]. Thus, dividing the group among their known conspecifics most probably did not represent a stressor for the animals.

An increasing number of studies reveals only few effects of the various aspects of group housing on the level of injuries, but rather show the beneficial effects of this housing form on welfare and health of horses [16, 19, 20, 53, 54]. However, it cannot be ruled out that under other conditions, fission and fusion of groups may resemble a stressor in horses. Many studies in various species have so far shown that the occurrence or strength of agonistic interactions depends on many factors, one of which among horses seems to be a training in appropriate behaviors [20, 55]. Other relevant factors for agonistic interactions are group stability, group composition with regard to sex and age, individual coping styles, experience and individual behavior characteristics of group members [7, 19, 20, 56]. Certain conditions and group structures might thus favor aggressive encounters, and in consequence increase stressor burdens within group-housed horses. Future studies specifically focusing on immune parameters should investigate whether these different scenarios lead to acute or long-lasting immune alterations.

In conclusion, relocation to individual stabling represented an intense stressor for the horses of the present study, leading to acute and lasting alterations in blood counts of various leukocyte types. In contrast, fission of the stable group did not result in behavioral, endocrine or immunological stress responses by the horses. The results of the present study therefore strongly indicate that social isolation is a chronic stressor with negative impact on welfare and health of horses and highlight the advantage of group housing systems in view of immunocompetence. Nevertheless, future studies should examine the effects of particular housing conditions on the immune system and resulting immunocompetence of horses. The results could help making improvements to management practices in order to increase health and welfare of domestic horses.

## Acknowledgments

The authors thank the veterinarians and the employees of the State Stud Farm Marbach for blood sampling, for their time and efforts during the experiments, and for their excellent animal care. We greatly appreciate assistance in the laboratory by P. Veit, F. Haukap, L. Reiske, C. Heyer, and C. Schalk. We also thank J. Hartung for statistical advice and C. Frasch for proofreading of the manuscript.

## Author Contributions

**Conceptualization:** Sonja Schmucker, Konstanze Krüger, Volker Stefanski.

**Data curation:** Sonja Schmucker.

**Formal analysis:** Sonja Schmucker, Vanessa Preisler.

**Funding acquisition:** Konstanze Krüger, Volker Stefanski.

**Investigation:** Sonja Schmucker, Vanessa Preisler, Isabell Marr.

**Methodology:** Sonja Schmucker, Konstanze Krüger.

**Project administration:** Sonja Schmucker, Isabell Marr, Konstanze Krüger.

**Validation:** Sonja Schmucker, Konstanze Krüger.

**Visualization:** Sonja Schmucker.

**Writing – original draft:** Sonja Schmucker.

**Writing – review & editing:** Isabell Marr, Konstanze Krüger, Volker Stefanski.

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
