## [Decision Letter · Decision Letter 0]

6 Jun 2022

PONE-D-22-07855Single housing but not changes in group composition causes stress-related immunomodulations in horsesPLOS ONE

Dear Dr. Schmucker,

Thank you for submitting your manuscript to PLOS ONE. After careful consideration, we feel that it has merit but does not fully meet PLOS ONE’s publication criteria as it currently stands. Therefore, we invite you to submit a revised version of the manuscript that addresses the points raised during the review process.

The Reviewers provided some very helpful suggestions that will improve the overall impact of your article.

We look forward to receiving your revised manuscript.

Kind regards,

Fulvio D'Acquisto, PhD

Academic Editor

PLOS ONE

Journal Requirements:

Reviewers' comments:

Reviewer's Responses to Questions

**Comments to the Author**

1. Is the manuscript technically sound, and do the data support the conclusions?

Reviewer #1: Yes

Reviewer #2: Yes

2. Has the statistical analysis been performed appropriately and rigorously? 

Reviewer #1: Yes

Reviewer #2: Yes

3. Have the authors made all data underlying the findings in their manuscript fully available?

Reviewer #1: Yes

Reviewer #2: Yes

4. Is the manuscript presented in an intelligible fashion and written in standard English?

Reviewer #1: Yes

Reviewer #2: Yes

5. Review Comments to the Author

Reviewer #1: the study is interesting and well conducted. my comments are :

- in figure 2 in all the plots it would be better to indicate the identified populations instead of naming them with A B 1,2, etc. etc.

-it would also be advisable to calculate and show the neutrophil to lymphocyte ratio (NLR), and comment on the results obtained.

-it would be interesting to understand if the results obtained and shown in figures 2 and 3 are related in some way to the cortisol levels that have been observed

Reviewer #2: Comments to author

Generally, a well written manuscript. Please see my comments regarding the abstract. Otherwise minor grammatical edits and subtle changes or clarifications required. Discussion is balanced and fair.

Specific comments

Abstract

I would like to see the abstract rewritten. There is a considerable introduction component to the abstract and a lack of metrics to support statements made. Many readers will just skim the abstract and there is a need to provide some metrics to permit readers to be able to judge from the metrics in the abstract the relative strength of the effect.

Introduction

Line 56 change mostly to most

line 67 …in horses as indicated by…….

Line 68 please change heart activity to heart rate as ref 13 measured heart rate , 14 and 15 did not measure HRV so can be specific – heart rate

Line 85 please consider rewriting this sentence as it is grammatically clumsy - this was an intervention study so more appropriate to describe a large % of horses were outside the reference range – rather than prevalence

Materials and methods.

Line 115 – consider adding …without visual contact, between groups.

Line 145 used to is colloquial consider, …….. already familiarized to being guided……

Line 147 …venipuncture…. Rather than puncture

Line 155 consider replacing determined with measured or quantified

Line 242 given plosone has a broad readership please consider 1 or 2 lines to provide a bit more detail / elaboration here for readers not overtly familiar with behaviour measurement techniques.

Line 243 was any observation made of stereotypic behaviours of the horses when at pasture. If not then needs a statement here to justify why not measured.

Line 288 tree fell in the paddocks

Line 429 …in single boxes within one week….

Line 430 not sure what you mean by partly weaving – you have an ethogram and so to be precise this horse either is weaving or not .

Line 433 replace like with …..such as….

6. PLOS authors have the option to publish the peer review history of their article (what does this mean?). If published, this will include your full peer review and any attached files.

Reviewer #1: **Yes: **Enrico Gugliandolo

Reviewer #2: No

---

## [Author Response · Author response to Decision Letter 0]

18 Jul 2022

Responses to reviewer comments

Manuscript Number: PONE-D-22-07855 

Title: Single housing but not changes in group composition causes stress-related immunomodulations in horses

We thank the reviewers for their valuable and constructive comments, which helped us to improve the manuscript. We have revised the paper according to the suggestions and commented on all points mentioned. 

All changes are marked-up by the “track changes” option in the text of the manuscript named 'Revised Manuscript with Track Changes'. In addition, new line numbers within the unmarked version of the revised paper without tracked changes (named 'Manuscript') are specified within this rebuttal letter. 

Journal Requirements:

Response: We followed the style requirements named in the PLOS One style template in conjunction with the submission guidelines on the PLOS One website (https://journals.plos.org/plosone/s/submission-guidelines). Unfortunately, there are some discrepancies regarding the style requirements for the main text body. According to the submission guidelines on the PLOS One website they should be double-spaced, whereas in the style templates they are single-spaced and intended. We decided to prepare the manuscript with double-spaced text to achieve optimal readability. All Figure citations are now corrected (line 113, 318, 345 of the revised manuscript). 

Reviewer: 1

Reviewer #1: the study is interesting and well conducted. my comments are:

• in figure 2 in all the plots it would be better to indicate the identified populations instead of naming them with A B 1,2, etc. etc.

Response: We followed your suggestion and named the populations directly in the figures. (see revised Fig2.tiff). Accordingly, we also changed the corresponding figure captions (line 185 in the revised form of the manuscript).

• it would also be advisable to calculate and show the neutrophil to lymphocyte ratio (NLR), and comment on the results obtained.

Response: NLR as suggested is now provided. We agree that this is a more commonly used measure and thus will increase comparability to other studies. We skipped on the other hand the neutrophil-to-T cell ratio as changes in N:T ratio parallel the changes in NLR to a very high degree in this study. Accordingly, we replaced the N:T ratio in figure 3 by the NLR (see revised Fig3.eps), adjusted the figure caption (lines 354f of the revised manuscript), the corresponding part in material and methods (lines 178f, 287 of the revised manuscript), as well as the results paragraph (lines 316, 328ff, 344f, 347, 349 of the revised manuscript).

• it would be interesting to understand if the results obtained and shown in figures 2 and 3 are related in some way to the cortisol levels that have been observed

Response: We agree that this aspect could be of potential interest but decided not to go into much detail. The reason is that we did not find any correlation of cortisol and immune cell data. The more, when defining the best fitting model, we included cortisol as a covariate in the LMM and also separately analyzed both treatments: If only cortisol is included, a significant association is found for all investigated immune cell types within the treatment “relocation to individual housing”, which shows, that the change in cortisol concentrations indeed explains some of the variation found in immune cell counts. However, as soon as the fixed effect time or the interactive effect of treatment x time is included in the model, cortisol is no longer found to be associated with immune cell counts, showing that the effects time and treatment better account for the found differences in cell counts than cortisol. These statistical results reflect the fact, that cortisol raises only temporary (day one after relocation) but return to baseline level within the following week (day eight after relocation), whereas blood counts of immune cells show long-lasting modulations. We think that a detailed discussion of this complex statistical matter would not add much information to the manuscript but rather reduce clarity. It should be mentioned that we already discuss the results regarding cortisol levels based on the presented findings and cite other studies with comparable results in the discussion paragraph of the manuscript. We thereby conclude that immune cell counts might be a more suitable marker of stress responses (especially chronic stress responses) than cortisol concentrations. To increase clarity of this point, we carefully rephrased the according passage (line 481-486 in the revised form of the manuscript).

Reviewer: 2

Reviewer #2: Comments to author

Generally, a well written manuscript. Please see my comments regarding the abstract. Otherwise minor grammatical edits and subtle changes or clarifications required. Discussion is balanced and fair.

Specific comments

Abstract

• I would like to see the abstract rewritten. There is a considerable introduction component to the abstract and a lack of metrics to support statements made. Many readers will just skim the abstract and there is a need to provide some metrics to permit readers to be able to judge from the metrics in the abstract the relative strength of the effect.

Response: Thank you for this valuable hint. We adjusted the abstract according to your suggestions (line 26-47 of the revised manuscript). We also included the metrics named in the abstract in the results part to increase conformity (line 321ff, 332ff in the revised form of the manuscript).

Introduction

• Line 56 change mostly to most

Response: Changed (line 56 in the revised form of the manuscript)

• Line 67 …in horses as indicated by…….

Response: Changed (line 67 in the revised form of the manuscript)

• Line 68 please change heart activity to heart rate as ref 13 measured heart rate, 14 and 15 did not measure HRV so can be specific – heart rate

Response: Thank you for this hint. Sentence changed according to your suggestion (line 68 in the revised form of the manuscript).

• Line 85 please consider rewriting this sentence as it is grammatically clumsy - this was an intervention study so more appropriate to describe a large % of horses were outside the reference range – rather than prevalence

Response: Sentence changed to “Lesimple et al. (2020) found that blood cell counts of leukocytes fall outside the normal range within a high percentage of single-housed horses [25].” (line 85ff in the revised form of the manuscript)

Materials and methods.

• Line 115 – consider adding …without visual contact, between groups.

Response: Thank you for this suggestion. We added “[…] and did not allow visual contact between the groups.” to clarify this point. (line 115 in the revised form of the manuscript)

• Line 145 used to is colloquial consider, … already familiarized to being guided……

Response: We changed the word “used” into “habituated” to meet your suggestion. (line 146 in the revised form of the manuscript)

• Line 147 …venipuncture…. Rather than puncture

Response: changed (line 148 in the revised form of the manuscript)

• Line 155 consider replacing determined with measured or quantified

Response: changed to “quantified” (line 156 in the revised form of the manuscript)

• Line 242 given plosone has a broad readership please consider 1 or 2 lines to provide a bit more detail / elaboration here for readers not overtly familiar with behaviour measurement techniques.

Response: We took up your advice and included some more details on the techniques by adding “[…] by observing all individuals simultaneously for displaying one of the above-named agonistic and affiliative behaviors (behavioral sampling) and recording each occurrence as well as sender and receiver of the particular behaviors (continuous recording) […]” to the sentence in line 241 (line 245-248 in the revised manuscript). 

We also changed the sentence in line 244 to “Observations were made once per horse for 2 hours by continuous recording and behavioral sampling of the behaviors named in Table 1 by focusing on five to six animals in parallel (focal sampling) [33] by one experienced observer.“ to also increase clarity of the focal sampling technique (line 254f in the revised manuscript).

• Line 243 was any observation made of stereotypic behaviours of the horses when at pasture. If not then needs a statement here to justify why not measured.

Response: Preliminary observations did not reveal any signs of stereotypic behaviors shown by the horses while on pasture. Based on literature we also did not expect stereotypic behavior after changes in the group composition as usually the number, pattern, strength as well as duration of agonistic interactions might be indicative of a potential stress burden in group housed horses on pasture. We therefore, did not include stereotypic behavior patterns but rather social interactions in the ethogram for characterization of probable stress burden in the horses when prone to changes in group composition. In contrast, the occurrence of stereotypic behaviors is already described and discussed as suitable indicator of stress perception by (single) stabled horses. Thus, the ethograms for behavioral evaluation of probable stress responses by the horses were different for the two treatments. To clarify this point and increase soundness for readers we added this background information in the respective paragraphs of the material and methods section (lines 216-222 and 249-251 in the revised manuscript.)

• Line 288 tree fell in the paddocks

Response: changed (line 294 in the revised manuscript)

• Line 429 …in single boxes within one week….

Response: We are not sure whether “within” is grammatically right in conjunction with the verb “exhibit”. We could change the sentence to “the horses developed stereotypies within one week.”, but as we did not measure behavior before day 8 after relocation we would prefer to stay with the objective statement that we found the horses performing stereotypies in the second week after they had been relocated to single stabling. We therefore also deleted the word “already” (line 435 in the revised form of the manuscript)

• Line 430 not sure what you mean by partly weaving – you have an ethogram and so to be precise this horse either is weaving or not.

Response: Thank you for this hint. The horse did not show the whole weaving circle, but showed parts of this behavior. However, as you mention, this is defined in the ethogram. Therefore, we deleted “partly” in this sentence. (line 436 in the revised manuscript)

• Line 433 replace like with …..such as….

Response: changed (line 439 in the revised manuscript)

---

## [Editor Report · Decision Letter 1]

20 Jul 2022

Single housing but not changes in group composition causes stress-related immunomodulations in horses

PONE-D-22-07855R1

Dear Dr. Schmucker,

We’re pleased to inform you that your manuscript has been judged scientifically suitable for publication and will be formally accepted for publication once it meets all outstanding technical requirements.

Kind regards,

Fulvio D'Acquisto, PhD

Academic Editor

PLOS ONE

Additional Editor Comments (optional):

The authors have sufficiently addressed all the concerns raised by the Reviewers.
---

## [Editor Report · Acceptance letter]

22 Jul 2022

PONE-D-22-07855R1 

Single housing but not changes in group composition causes stress-related immunomodulations in horses 

Dear Dr. Schmucker:

I'm pleased to inform you that your manuscript has been deemed suitable for publication in PLOS ONE. Congratulations! Your manuscript is now with our production department. 

Kind regards, 

on behalf of

Professor Fulvio D'Acquisto 

Academic Editor

PLOS ONE